# Data-Efficient Instance Generation from Instance Discrimination

**Ceyuan Yang**[†]    **Yujun Shen**[‡]    **Yinghao Xu**[†]    **Bolei Zhou**[†]
[†]The Chinese University of Hong Kong    [‡]ByteDance Inc.

## Abstract

Generative Adversarial Networks (GANs) have significantly advanced image synthesis, however, the synthesis quality drops significantly given a limited amount of training data. To improve the data efficiency of GAN training, prior work typically employs data augmentation to mitigate the overfitting of the discriminator yet still learn the discriminator with a bi-classification (*i.e.*, real *vs.* fake) task. In this work, we propose a data-efficient Instance Generation (*InsGen*) method based on instance discrimination. Concretely, besides differentiating the real domain from the fake domain, the discriminator is required to distinguish every individual image, no matter it comes from the training set or from the generator. In this way, the discriminator can benefit from the infinite synthesized samples for training, alleviating the overfitting problem caused by insufficient training data. A noise perturbation strategy is further introduced to improve its discriminative power. Meanwhile, the learned instance discrimination capability from the discriminator is in turn exploited to encourage the generator for diverse generation. Extensive experiments demonstrate the effectiveness of our method on a variety of datasets and training settings. Noticeably, on the setting of $2K$ training images from the FFHQ dataset, we outperform the state-of-the-art approach with 23.5% FID improvement.[1]

## 1  Introduction

Generative Adversarial Network (GAN) [16] has become a popular paradigm to learn the distribution of the observed data. It is formulated as a two-player game, where a generator synthesizes realistic data, while a discriminator distinguishes synthesized samples from real ones. To reach equilibrium in this minimax game, it requires both the generator and the discriminator to be sufficiently trained. In other words, the synthesis capability of the generator will subsequently deteriorate given an inadequate discriminator [24, 39, 49, 51].

Recent success of GANs [22, 23, 25, 4] relies on big data to assure the sufficient training of the discriminator. Prior work [49, 24] has found that reducing the amount of training data leads to the overfitting of the discriminator, which tends to memorize the entire training set. In turn, the back-propagation from the discriminator to the generator damages the synthesis quality of the generator and potentially causes the mode collapse problem [1, 44]. Data augmentation is one of the most widely used methods to alleviate the overfitting issue in deep learning algorithms [45, 11, 10]. Some recent attempts [24, 39, 49, 51, 44] have been made to apply data augmentation to GAN training. It is found that the discriminator can be improved by augmenting not only the real images from the dataset but also the synthesized images by the generator [49, 24]. However, the learning objective of the discriminator remains as categorizing real and fake domains and a substantial performance drop can be observed given limited training data.

---

[1]Code is available at https://genforce.github.io/insgen/.

35th Conference on Neural Information Processing Systems (NeurIPS 2021).

The domain bi-classification task could be too easy for the discriminator to gain sufficient discriminative power as an adaptive loss to train the generator, especially when the size of training set is small. In this work, we propose to improve the data efficiency in GAN training by assigning a more challenging task to the discriminator, which is to distinguish every individual image as an independent category. In this way, the discriminator is forced to improve its discriminative capability to accomplish the instance discrimination task [40]. Notably, besides distinguishing real samples, we also demand the discriminator to differentiate fake samples synthesized by the generator. Thus the discriminator can be considered to train with infinite data, preventing it from memorizing the training samples. When distinguishing synthesized data, we design a noise perturbation strategy to increase the difficulty of the task and hence make the discriminator more capable. Meanwhile, we also alter the training objectives from the generator side. Concretely, besides making the generator to fool the discriminator, we expect all the samples produced by the generator to be well identified as different instances with our instance-induced discriminator. This highly matches the goal of diverse generation, which requires every synthesis to be unique. We evaluate our method on a range of datasets and achieve appealing generation performance in terms of image quality, diversity, and data efficiency. Experiments show that our method significantly improves the baselines and outperform previous data-augmentation methods. To be specific, our method improves the FID from 15.60 to 11.92, 7.29 to 4.90, and 3.88 to 3.31 with $2K$, $10K$, and $70K$ training images from FFHQ [23] respectively. We can even learn a large-scale GAN with only 100 in-the-wild images to produce satisfying synthesis.

Our main **contributions** are summarized as follows: 1) We propose a data-efficient instance generation (*InsGen*) method which incorporates instance discrimination as an auxiliary task in GAN training. 2) The synthesized data is used as infinite samples for improving the discriminative power of the discriminator, which in turn substantially improves the synthesis quality and diversity of the generator. 3) Under various data-regime settings, our method consistently surpasses existing alternatives by a substantial margin.

## 2   Related Work

**Data Augmentation in GANs.** Data augmentation makes the maximum use of available data to alleviate the overfitting of deep models that have millions of parameters. It plays an essential role in training discriminative models [45, 11, 10]. Some recent work explores how data augmentation can help the training of GANs [51, 39, 49, 24]. Zhao et al. [51] conduct empirical studies on the effects of different types of augmentations for GAN training. Tran et al. [39] make a theoretical analysis of several data augmentations. Zhao et al. [49] propose a differentiable augmentation method such that the augmenting operations can be applied to both real and synthesized data. Similarly, Karras et al. [24] design augmentations that do not leak and introduce a probability-based adaptive strategy to stabilize the training process. Different from prior work, we focus on introducing the unsupervised representation learning which also requires augmentations into GAN training. Our work shows that the recent instance discrimination task [40] can be used as an auxiliary task for the discriminator, which in turn substantially improves the synthesis quality of the generator.

**Self-supervised Learning in GANs.** The rationale behind self-supervised learning is to set up various pretext tasks with supervisory-free labels [14, 5, 41, 48, 13, 32, 34, 42, 34, 15, 31, 35]. Similar idea is recently introduced in GAN training as an auxiliary loss to improve the synthesis performance. For instance, Chen et al. [6] assign the rotation prediction task to the discriminator to prevent it from catastrophic forgetting, and Tran et al. [38] propose a multi-class minimax game to encourage the generator to produce diverse samples. Among all self-supervised learning approaches, contrastive learning [40, 17, 7, 18, 3] shows great potential in large-scale representation learning. Many attempts have been made to improve generative models by drawing lessons from contrastive learning, like the consistency regularization for GANs [47, 50], the patch-level contrastive learning for image-to-image translation [33], and the latent-augmented contrastive loss for conditional image synthesis [29]. Akin to supervised contrastive loss [27], some concurrent work [20, 21, 43] reformulates the conventional bi-classification task (*i.e.*, real domain *vs.* fake domain) with contrastive loss. Differently, we keep the original bi-classification task of the discriminator and introduce contrastive learning as a new one. Specifically, we assign the discriminator a simple auxiliary task, which is to *recognize every individual image*, no matter it is real or synthesized by the generator. Such instance discrimination task helps sustain the discriminative power of the discriminator under a low-data regime, which in turn improves the synthesis performance significantly.

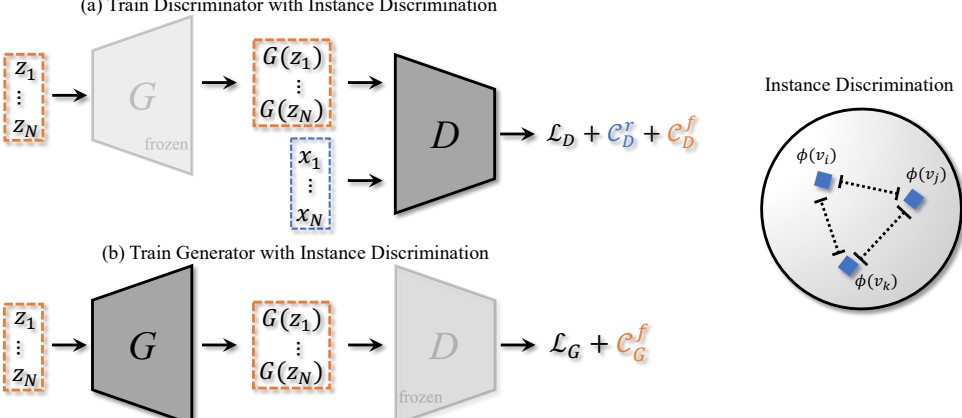

Figure 1: Illustration of the *InsGen* method. Besides the bi-classification task to differentiate real and fake domains, the discriminator is assigned an auxiliary task, which aims at maximally distinguishing each image instance as illustrated on the right. $\mathcal{C}$ denotes the training objective for such instance discrimination task. (a) The discriminator is asked to recognize not only every real sample $\mathbf{x}_i$ but also every synthesized sample $G(\mathbf{z}_i)$ by a frozen generator. (b) With the instance-induced discriminator, the generator is encouraged to make all synthesis recognizable from each other, leading to more diverse generation.

## 3 Methodology

In this section, we introduce the proposed InsGen method. Recall that our method is built based on GAN, which is commonly formulated as a two-player game between a generator and a discriminator. They compete with each other in that the generator tries to produce as realistic data as possible while the discriminator works on recognizing synthesized data from real data. Besides the conventional bi-classification task (*i.e.*, differentiating real and fake domains), we also require the discriminator to *distinguish every individual instance*. With such a challenging task, the discriminator can mitigate the overfitting problem even with limited training data. We will briefly introduce the image synthesis and instance discrimination mechanisms in Sec. 3.1, followed by our improved training pipeline in Sec. 3.2 and the practical usage of InsGen on the state-of-the-art StyleGAN2-ADA model [24] in Sec. 3.3.

### 3.1 Preliminaries

Our work is highly related to GAN [16] for image synthesis and contrastive learning [40, 17] for instance discrimination. To make the paper self-contained, we shortly describe these two algorithms in the text below.

**Synthesizing Images with GANs.** GAN is a popular paradigm for image generation. It typically consists of two networks: a generator $G(\cdot)$ that learns to map a latent variable $\mathbf{z}$ to a photo-realistic image, and a discriminator $D(\cdot)$ that aims at separating real images $\mathbf{x}$ from synthesized ones $G(\mathbf{z})$. These two networks compete with each other [16] and are jointly optimized with

$$\mathcal{L}_D = -\mathbb{E}_{\mathbf{x}\in\mathcal{X}}[\log(D(\mathbf{x}))] - \mathbb{E}_{\mathbf{z}\in\mathcal{Z}}[\log(1 - D(G(\mathbf{z})))], \tag{1}$$
$$\mathcal{L}_G = -\mathbb{E}_{\mathbf{z}\in\mathcal{Z}}[\log(D(G(\mathbf{z})))], \tag{2}$$

where $\mathcal{Z}$ and $\mathcal{X}$ denote the pre-defined latent distribution and real data distribution respectively. After the training converges, the synthesized images are assumed to be as realistic as real ones to fool the discriminator. From this perspective, the synthesis quality highly depends on the discriminative power of the discriminator. Prior literature [24, 39, 49, 51] has affirmed that GANs will suffer from the insufficient training of the discriminator and proposed to apply a series of data augmentations $\mathcal{T}(\cdot)$ to alleviate the overfitting problem. But they do not change the learning objectives of GAN and observe drastic performance drop given limited training data.

**Distinguishing Images with Contrastive Learning.** It is well-known that image classification tasks usually benefit from more discriminative representations [12]. Unlike supervised training algorithms

that optimize the model parameters based on annotated data, contrastive learning [40, 17, 7, 18, 3] is able to extract representative features from images in an unsupervised manner. As shown in Fig. 1a, the rationale behind is to "label" every sample as an individual class, *i.e.*, instance discrimination. Concretely, given an image $\mathbf{x}$, two random "views" (*e.g.*, through different augmentations) are created as the query $\mathbf{x}_q$ and the key $\mathbf{x}_{k_+}$. This query-key pair is regarded as the positive pair while all "views" from other images, $\{\mathbf{x}_{k_i}\}_{i=1}^N$, are treated as negative pairs with respect to the query. Here, $N$ is the total number of images in addition to the query image. Contrastive learning aims at maximizing the agreement across augmentations (*i.e.*, $\mathbf{x}_q$ and $\mathbf{x}_{k_+}$) and make the query as much dissimilar to a number of negative samples as possible. Accordingly, we can design a pretext task of $(N+1)$-way classification and learn the model with the contrastive loss $\mathcal{C}$ *i.e.*, InfoNCE loss [32]

$$\mathbf{v}_q = F(\mathbf{x}_q), \quad \mathbf{v}_{k_+} = F(\mathbf{x}_{k_+}), \quad \mathbf{v}_{k_i} = F(\mathbf{x}_{k_i}), \ i = 1 \ldots N, \tag{3}$$

$$\mathcal{C}_{F(\cdot),\phi(\cdot)}(\mathbf{x}_q, \mathbf{x}_{k_+}, \{\mathbf{x}_{k_i}\}_{i=1}^N) = -\log \frac{\exp(\phi(\mathbf{v}_q)^T \phi(\mathbf{v}_{k_+})/\tau)}{\sum_{i=0}^N \exp(\phi(\mathbf{v}_q)^T \phi(\mathbf{v}_{k_i})/\tau)}, \tag{4}$$

where $F(\cdot)$ is the backbone network to extract the representation $\mathbf{v}$ from a given image $\mathbf{x}$, and $\phi(\cdot)$ is the head network (*e.g.*, usually implemented with several fully-connected layers) to project the extracted feature onto a unit sphere. $\tau$ stands for the temperature, which is a hyper-parameter. Recall that the primitive goal of the discriminator in GANs can also be viewed as a bi-classification task, which is to recognize real and fake domains. In this work, we demonstrate that introducing the instance discrimination task can help enhance the discriminative power of the discriminator and in turn improve the synthesis quality of the generator significantly.

## 3.2 Generating Diverse Instances from Distinguishing Instances

In this part, we will introduce how instance discrimination is incorporated into the GAN training for data-efficient and diverse image generation. There are four essential components of our InsGen method: 1) distinguishing real images, 2) distinguishing fake images that can be sampled infinitely, 3) a noise perturbation strategy, and 4) a loop-back mechanism to encourage the generator for the diverse generation.

**Distinguishing Real Images.** As discussed above, the synthesis quality of GAN models not only depends on the training scheme [2, 30, 22, 4] and the architecture design of the generator [46, 23, 25], but more importantly relies on the discriminative capability of the discriminator. That is because the discriminator is the only one (compared to the generator) that can see how real data looks like and further guides the generator accordingly. To make the maximum use of the limited training data and avoid the discriminator from memorizing the entire dataset, we assign it with a more challenging task beyond domain classification, which is to recognize every independent instance from the dataset, as shown in Fig. 1a. For this purpose, we introduce a new task head $\phi^r(\cdot)$ beyond the original bi-classification head $\phi^{domain}(\cdot)$ on top of its backbone $d(\cdot)$[2] and train the discriminator with an extra training objective

$$\mathcal{C}_D^r = \mathcal{C}_{d(\cdot),\phi^r(\cdot)}(\mathcal{T}_q(\mathbf{x}_q), \mathcal{T}_{k_+}(\mathbf{x}_q), \{\mathcal{T}_{k_i}(\mathbf{x}_{k_i})\}_{i=1}^N). \tag{5}$$

Here, $\mathbf{x}_q, \{\mathbf{x}_{k_i}\}_{i=1}^N$ are all sampled from the real data distribution $\mathcal{X}$ and transformed with various differentiable augmentations $\mathcal{T}(\cdot)$.

**Distinguishing Fake Images.** However, the amount of training data could be extremely few (like thousands or even hundreds) in practice. In such a case, the improvement of the discriminator gained by differentiating real instances will be also limited. On the other hand, we notice that the number of synthesized samples can be sufficiently large due to the sampling mechanism of GANs. Ideally, different latent codes $\mathbf{z} \in \mathcal{Z}$ should lead to different synthesis $G(\mathbf{z})$. Hence, we propose to also ask the discriminator to recognize every individual fake images, as shown in Fig. 1a. Similarly, we introduce another task head $\phi^f(\cdot)$ into the discriminator. It is worth mentioning that we use separate task heads (*i.e.*, $\phi^r(\cdot)$ and $\phi^f(\cdot)$) for real and fake data. That is because even though the synthesized images can be with high-quality, they still lie in a different distribution from the real ones, especially when the generator starts training from scratch. Meanwhile, the task of discriminating a real instance from a fake instance can be achieved by the native domain classification head $\phi^{domain}(\cdot)$.

---

[2]The conventional discriminator is a composition of $d(\cdot)$ and $\phi^{domain}(\cdot)$ to perform real/fake classification, *i.e.*, $D(\cdot) = \phi^{domain}(\cdot) \circ d(\cdot)$.

**Noise Perturbation.** Prior work has observed the continuity of the latent space [36] such that images synthesized from the latent codes within a neighbourhood are very close to each other. Accordingly, they are more suitable to be treated as positive pairs than negative pairs. From this perspective, we introduce a noise perturbation strategy into fake image discrimination. The objective becomes

$$\mathbf{x}'_q = \mathcal{T}_q(G(\mathbf{z}_q)), \quad \mathbf{x}'_{k_+} = \mathcal{T}_{k_+}(G(\mathbf{z}_q + \epsilon)), \quad \mathbf{x}'_{k_i} = \mathcal{T}_{k_i}(G(\mathbf{z}_{k_i})), \tag{6}$$

$$\mathcal{C}_D^f = \mathcal{C}_{d(\cdot),\phi^f(\cdot)}(\mathbf{x}'_q, \mathbf{x}'_{k_+}, \{\mathbf{x}'_{k_i}\}_{i=1}^N). \tag{7}$$

Concretely, given a query image $\mathbf{x}'_q$, the key image $\mathbf{x}'_{k_+}$ is created with $\mathcal{T}_{k_+}(G(\mathbf{z}_q + \epsilon))$ instead of $T_{k_+}(G(\mathbf{z}_q))$. Here, $\epsilon$ stands for the perturbation term, which is sampled from a Gaussian distribution whose variance is sufficiently smaller than that of $\mathcal{Z}$, and $\mathcal{T}_q(\cdot)$ and $\mathcal{T}_{k_+}(\cdot)$ denote two different augmentations. Such design aims to enforce the discriminator invariant to the small perturbation, which makes the instance discrimination task more challenging.

**Toward Diverse Generation.** Besides utilizing the instance discrimination task to improve the discriminative power of the discriminator, we further design a loop-back mechanism to in turn use the learned instance discrimination to guide the generator. Recall that image diversity, in addition to image quality, is also an important metric to evaluate generative models. Diverse generation, which requires all generated samples to be distinguishable from each other, exactly matches our goal of instance discrimination. In other words, given a discriminator with the ability to distinguish different instances, we would like all the samples produced by the generator to be recognized as different ones. This idea is illustrated in Fig. 1b. By comparing Fig. 1a and Fig. 1b, we can see that the generator shares the same target as the discriminator yet is trained separately. Hence, the same objective function is added into the generator loss

$$\mathbf{x}''_{k_+} = \mathcal{T}_{k_+}(G(\mathbf{z}_q)), \tag{8}$$

$$\mathcal{C}_G^f = \mathcal{C}_{d(\cdot),\phi^f(\cdot)}(\mathbf{x}'_q, \mathbf{x}''_{k_+}, \{\mathbf{x}'_{k_i}\}_{i=1}^N), \tag{9}$$

where the only difference is that noise perturbation is not applied during the training of the generator.

**Complete Objective Function.** To summarize, with the purposes of both image synthesis and instance discrimination, the discriminator and the generator in InsGen are optimized with

$$\mathcal{L}'_D = \mathcal{L}_D + \lambda_D^r \mathcal{C}_D^r + \lambda_D^f \mathcal{C}_D^f, \tag{10}$$

$$\mathcal{L}'_G = \mathcal{L}_G + \lambda_G \mathcal{C}_G^f, \tag{11}$$

where $\lambda_G$, $\lambda_D^r$, and $\lambda_D^f$ denote the weights for different terms.

### 3.3 Implementation

On top of the adversarial training pipeline in GANs, our InsGen method only inserts an extra loss output on the discriminator network for instance discrimination. Therefore, it can be easily implemented on any GAN framework. In this part, we take the state-of-the-art GAN model, StyleGAN2-ADA [24], as an example to demonstrate how InsGen is implemented in practice.

**Generative Model.** StyleGAN2-ADA [24] adopts the architecture of StyleGAN2 [25] and proposes the adaptive discriminator augmentation strategy for training with limited data. In particular, it designs a differentiable augmentation pipeline, consisting of 18 transformations, as well as an adaptive hyper-parameter to control the strength of these augmentations. For a fair comparison, in this work, we exactly reuse the network structure, the augmentation pipeline, the adaptive strategy of the augmenting strength, and other hyper-parameters like batch size and learning rate.

**Instance Discrimination.** We reuse the backbone of the discriminator to perform instance discrimination, so that the extra computing load is extremely small and the training efficiency is barely affected. We treat the last fully-connected layer in the StyleGAN2-ADA discriminator as the domain-classification head $\phi^{domain}(\cdot)$, while all remaining layers serve as the backbone network $d(\cdot)$. The real instance discrimination head $\phi^r(\cdot)$ and the fake head $\phi^f(\cdot)$ are both implemented with 2 fully-connected layers, followed by $\ell_2$ normalization. Strictly following MoCo-v2 [8], an extra queue is employed for each task head to store the sample features to save computational cost. The number of samples in $\mathcal{L}_D^r$ and $\mathcal{L}_D^f$ is thus equal to the queue size, which usually contains around 5% data of the whole set. We also introduce the momentum encoder $D'$, whose parameters are updated with moving average scheme: $\Theta_{D'} \leftarrow \alpha\Theta_{D'} + (1-\alpha)\Theta_D$. Here, $\alpha = 0.999$ follows the same setting in MoCo-v2 [8]. The temperature $\tau$ in Eq. (4) is set as 2.

Table 1: **Performance on FFHQ.** FID (lower is better) is reported as the evaluation metric. "2K", "10K", and "140K" stand for the number of samples used for training, where "140K" horizontally flips the original FFHQ dataset (with 70K samples) to double the size of data. Results with $*$ are also achieved with horizontally flipped data, which are slightly better than those reported in [24]. Numbers in **blue** color indicate our improvements over the baseline [24].

| 256×256 Resolution | 2K | 10K | 140K |
|---|---|---|---|
| PA-GAN [44] | 56.49 | 27.71 | 3.78 |
| zCR [50] | 71.61 | 23.02 | 3.45 |
| Auxiliary rotation [6] | 66.64 | 25.37 | 4.16 |
| StyleGAN2 [23] | 78.80 | 30.73 | 3.66 |
| w/ Shallow mapping [24] | 71.35 | 27.71 | 3.59 |
| w/ Adaptive dropout [24] | 67.23 | 23.33 | 4.16 |
| w/ DiffAugment [49] | 24.32 | 7.86 | - |
| w/ ADA [24] | 15.60* | 7.29* | 3.88 |
| **InsGen** (Ours) | **11.92** (−3.68) | **4.90** (−2.39) | **3.31** (−0.57) |

# 4 Experiments

We evaluate the proposed InsGen method on multiple benchmarks. Sec. 4.1 presents the comparison to prior literature on both FFHQ [23] and AFHQ [9] datasets. Our InsGen substantially improves the baselines under multiple data-regime settings and outperforms previous data-augmentation approaches by a significant margin. Moreover, Sec. 4.2 provides a detailed ablation study to show the importance of each component. Lastly Sec. 4.3 discusses about the limitation of data-efficiency.

## 4.1 Main Results

**Datasets.** We evaluate our InsGen with a number of other approaches on FFHQ [23] and AFHQ [9] datasets. FFHQ contains unique 70,000 high-resolution images (1024×1024), with large variation regarding age, ethnicity, and background. All images of FFHQ are well aligned [26] and cropped. In order to conduct a fair comparison, we resize images to 256×256. For the experiments of limited data, we follow ADA [24] to collect a subset of training data by randomly sampling. Moreover, AFHQ consists of around 5000 images per category for dogs, cats, and wild life at 512×512 resolution. Each category is regarded as a dataset and thus we train a different network on each dataset.

**Training.** We implement our InsGen on the official implementation of StyleGAN2-ADA. The training regularization is preserved, including path length regularization, lazy regularization, and style mixing regularization. Moreover, all parameters share the same learning rate and the minibatch standard deviation layer is adopted at the end of the discriminator. Exponential moving average of generator weights, non-saturating logistic loss with $R_1$ regularization, and Adam optimizer [28] is also adopted. In particular, the coefficient of gradient penalty would be decreased correspondingly, according to the official implementation of ADA [24]. All the experiments are conducted on a server with 8 GPUs. Mixed-precision training is also used for faster training.

**Hyper-parameters.** Empirically, the loss weights $\lambda_G$, $\lambda_D^f$ and $\lambda_D^r$ are 0.1, 1.0 and 1.0 respectively. Besides, the training length is slightly different. For the experiments with less than 10K images, the total number of seen images is 10 million rather than 25 million adopted by ADA [24]. Meanwhile, we decrease the loss weight of the gradient penalty due to involving an extra supervision. For example, ADA [24] adopts 1.0 for original StyleGAN2 training while we use 0.8. We also found smaller loss weight of gradient penalty is beneficial to our InsGen on the less data, *e.g.*, 0.3 and 0.5 for 10K and 2K experiments respectively.

**Evaluation Metric.** We use Fréchet Inception Distance (FID) [19] as the metric for quantitative comparison metric since FID tends to reflect the human perception of synthesis quality. As mentioned in Heusel et al. [19], we always calculate the FID between 50,000 fake images and all training images, no matter how much data the training set contains. The official pre-trained Inception network is used to compute the FID.

Table 2: **Performance on AFHQ.** FID (lower is better) is reported as the evaluation metric. Numbers in **blue** color indicate our improvements over the baseline [24].

| 512×512 Resolution | Cat | Dog | Wild life |
|---|---|---|---|
| StyleGAN2 [23] | 5.13 | 19.4 | 3.48 |
| ContraD [20] | 3.82 | 7.16 | 2.54 |
| ADA [24] | 3.55 | 7.40 | 3.05 |
| **InsGen** (Ours) | **2.60** (−0.95) | **5.44** (−1.96) | **1.77** (−1.28) |

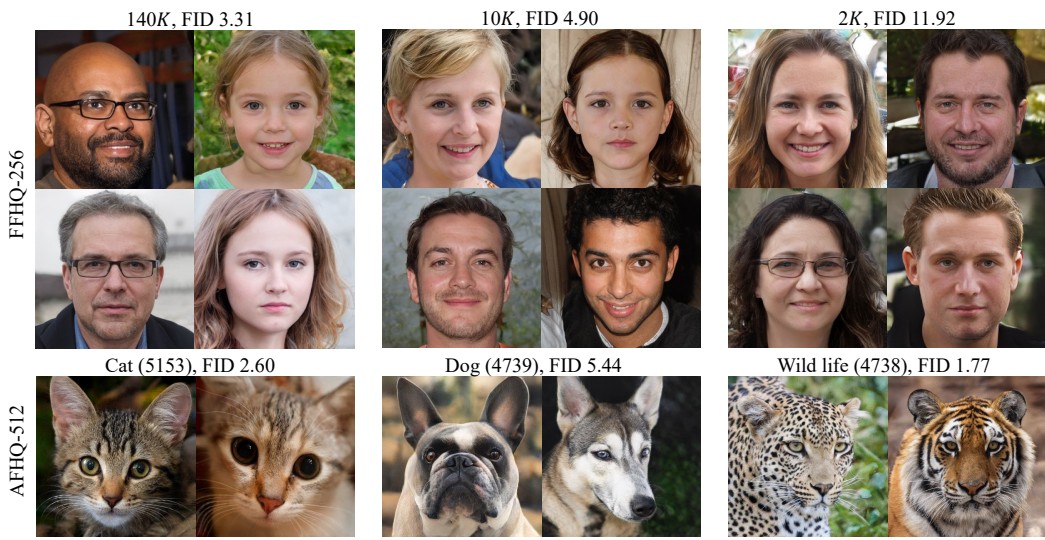

Figure 2: **Generated images under various data regimes.** The number of training images and the corresponding FID are reported. All images on FFHQ are synthesized with truncation following [24] while those on AFHQ are not.

**Results on FFHQ.** Tab. 1 presents the comparison on FFHQ. Akin to ADA [24], we compare against PA-GAN [44], zCR [50] and auxiliary rotation [6]. Also, StyleGAN2 together with its variants is also introduced as the baseline methods. For instance, less data is usually required when a shallower mapping network is applied. Besides, dropout [37] is also well-studied to be replaced with the augmentations as the regularization. Note that * means the dataset is amplified by 2× via the horizontal flip, which is recommended in the official implementation of ADA [24]. Such that, "2$K$" denotes 2,000 unique images and the dataset is enlarged to 4,000 via the flip operation, leading to a better baseline.

Although ADA [24] has already improved the performance significantly under various low-data regimes, our InsGen continues to improve the low-data image generation by a clear margin, establishing a new state-of-the-art synthesis quality with limited training images. To be specific, our method improves the FID from 15.60 to 11.92, 7.29 to 4.90, and 3.88 to 3.31 with 2$K$, 10$K$ and 70$K$ training images from FFHQ [23] respectively. Fig. 2 presents several generated examples under various data regimes. More qualitative results are available in our supplementary material. All images on FFHQ are generated with truncation. It is also worth noting that our approach further improves the synthesis quality when the full dataset is given, even outperforming previous best one *i.e.*, zCR [50]. Namely, the data can be further exploited when it is not the bottleneck for training.

**Results on AFHQ.** We also evaluate our approach on AFHQ dataset [9] which is divided into cat, dog and wild life, with the number of 5153, 4739 and 4738 images respectively. Therefore, three models are trained on them individually. Note that all models on AFHQ are trained on 512×512 images while the generated samples are resized to present. We involve StyleGAN2 [25], ContraD [20] and ADA [24] as the baseline approaches, compared to our InsGen. Quantitative and qualitative results are shown in Tab. 2 and Fig. 2 respectively.

The synthesis quality on those datasets is substantially improved by our method, which also outperforms previous data-augmentation methods. To be specific, our method improves the FID

Table 3: **Ablation Study.** FID (lower is better) is reported as the evaluation metric. Here, vanilla $\mathcal{C}_D^f$ means that the noise perturbation is not applied in the fake instance discrimination.

| $\mathcal{C}_D^r$ | vanilla $\mathcal{C}_D^f$ | $\mathcal{C}_D^f$ | $\mathcal{C}_G^f$ | $2K$ | $10K$ | $70K$ |
|:---:|:---:|:---:|:---:|:---:|:---:|:---:|
| | | | | 15.60 | 7.29 | 3.76 |
| ✓ | | | | 14.15 | 5.98 | 3.56 |
| ✓ | ✓ | | | 13.46 | 5.68 | 3.67 |
| ✓ | ✓ | ✓ | | 12.19 | 5.30 | 3.49 |
| ✓ | ✓ | ✓ | ✓ | 11.92 | 4.90 | 3.31 |

from 3.55 to 2.60, 7.40 to 5.44, and 3.05 to 1.77 on cat, dog and wild life images respectively. In particular, ContraD [20] introduced stronger augmentations to train a better discriminator via contrastive learning. One term in this method shares the similar motivation that real images could result in powerful representations. In terms of the use of synthesized samples, ContraD turned to focus on the binary classification, (*i.e.*, real *vs.* fake) with some specific designs like the stop-gradient operation. Differently, our method leverages the generated images as a kind of data complement to produce a stronger representation and guide the learning of the generator. Accordingly, InsGen achieves the new state-of-the-art performances on AFHQ [9].

## 4.2 Ablation Study

In order to investigate the importance of each component in our InsGen, we conduct an ablation study on FFHQ [23] with the image resolution of $256 \times 256$. FID serves as the main metric for the comparison, and the results on $2K$, $10K$ and $70K$ unique images are reported. During training each unique image go through random flip operation to obtain a stronger baseline. Tab. 3 presents the collection of various experiments in the ablation study. We choose the ADA [24] as the baseline.

**How important is the instance discrimination?** After performing the real image discrimination, the synthesis quality is improved, with the FID consistently decreased by **-1.45**, **-1.31** and **-0.20** in Tab. 3, no matter how many unique images the training set includes. To some extent, the discriminator would benefit from the powerful representations derived from the challenging pretext task. Accordingly, the generator is required to produce more photo-realistic images in order to confuse the discriminator.

When adding instance discrimination with fake images, performances could be further boosted. For instance, FID obtains an improvement of **-0.69** and **-0.30** with $2K$ and $10K$ images respectively. In particular, the gains rise as the number of real images goes down, verifying one of our motivations that the fake samples can be also regarded as data source for unsupervised representation learning.

**How important is the noise perturbation?** In Sec. 3.2, a noise perturbation strategy is proposed as a type of latent space augmentation for fake image discrimination. In particular, this latent space augmentation, *i.e.*, the small movement in the latent space always leads to an obvious but semantically consistent change of the original image, which could not easily be implemented by some geometric and color transformations. Meanwhile, the discriminator is required to be invariant to such noise perturbation due to the goal of instance discrimination. Accordingly, the fake images are made best use of to result in stronger representations for the discrimination. As shown in Tab. 3, such strategy further brings consistent gains of **-1.27**, **-0.38** and **-0.18** on $2K$, $10K$ and $70K$ datasets respectively.

**How important is the supervision signal for the generator?** The last row of Tab. 3 shows the performances with the gradients which are back-propagated to the generator. Even if we have already obtained quite strong results, such a supervision signal on the generator could also introduce improvements under various data regimes.

The goal of instance discrimination is to distinguish every individual image according to its appearance cues [40]. Assuming this pretext task is well-performed on a fixed dataset, the semantic representation would be derived from this learning process. However, when distinguishing fake images, the fake dataset actually varies dynamically. Namely, we could accomplish this pretext task from the perspective of data, if the engine of this dynamical fake dataset, *i.e.*, the generator could produce as many different images as possible. In general, this pretext task is exploited to encourage the diverse generation directly on the generator.

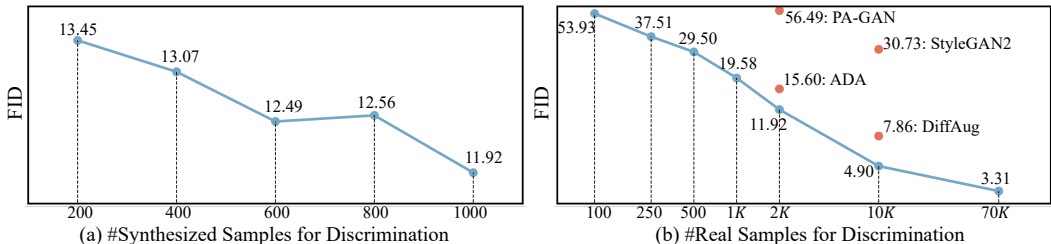

(a) #Synthesized Samples for Discrimination    (b) #Real Samples for Discrimination

Figure 3: **Effect of the number of synthesized and real images used for instance discrimination.** FID (lower is better) in log-scale is reported as the evaluation metric. We can see the consistent performance gain along with the increasing number of instances for discrimination.

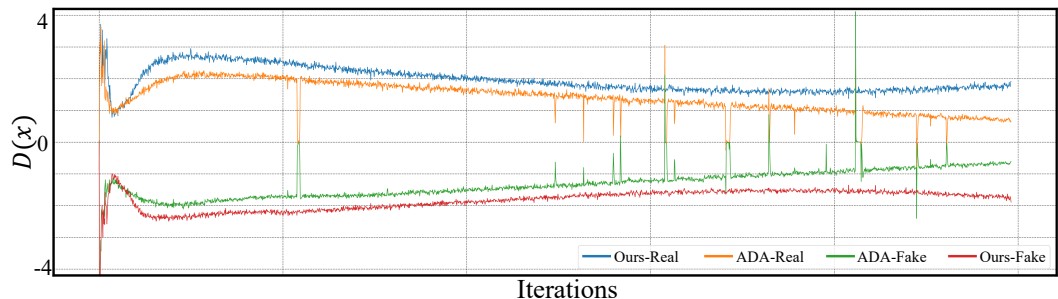

Iterations

Figure 4: Training progress on FFHQ-$2K$. Larger value means that the image is more realistic under the view of the discriminator. Our discriminator can *better and more stably* differentiate real and fake data compared to ADA [24].

**How important is the number of negative samples?** We follow the MoCo-v2 [8] to store multiple features in a queue, in order to reduce the computational complexity. Empirically, the length of the feature queue tends to be the 5% number of the dataset. Therefore, it is 200 when we have $2K$ unique images and enlarge them via the flip operation. However, there is no any reference number for the synthesized data. Accordingly, we collect as the same amount of fake data as that of the real.

As mentioned in Sec. 3.2, there could be much more synthesized samples than the real samples. Namely, we could leverage infinite samples for the synthesized instance discrimination. Therefore, we investigate the effect of the different number of synthesized samples *i.e.,* the length of the feature queue, shown in Fig. 3a. Obviously, FID gradually decreases with the increasing number of synthesized samples, suggesting that involving more fake images is of great benefit to the synthesis, especially with the limited training data.

**Whether the discriminative ability of the discriminator is really enhanced.** As is mentioned in our work, it is challenging to gain sufficient discriminative power for the discriminator to train the generator when the size of training set is small. However, introducing instance discrimination is able to improve its discriminative capability, achieving new state-of-the-art synthesis quality. In order to investigate whether the discriminative ability is improved, we plot the logits (derived from the discriminator) of any input image during the training in Fig. 4. To be specific, the logit denotes how much the input image is identified as the real. And the number of training images are 2000.

Obviously, our method produces higher real and lower fake scores throughout the whole training progress, compared to the baseline approach ADA [24]. It indicates that the discriminator of our method performs the domain bi-classification (*i.e.*, real *vs.* fake) better than that of baseline, showing stronger discriminative ability. It also verifies our motivation that a challenging pretext task which is to distinguish every individual image could indeed enhance the discriminator. Besides, the training progress is much more stable when equipped with our approach.

### 4.3 Towards the Limit of Data-efficiency

Although we have obtained the new state-of-the-art synthesis performances under the standard settings, we also wonder how much data-efficiency our InsGen could achieve. Therefore, the number

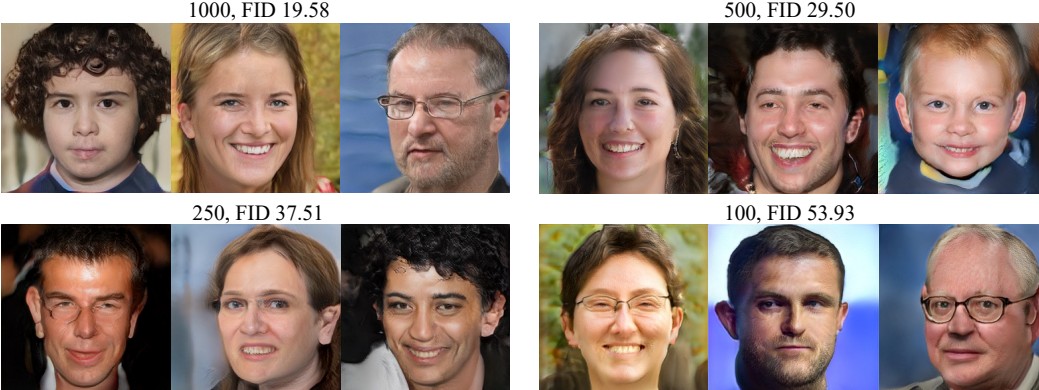

Figure 5: **Qualitative results with different number of training images.** The number of training images and the corresponding FID are reported. All images are synthesized with truncation following [24].

of real data in the training set is further reduced to 1000, 500, 250 and 100. In order to conduct the apple-to-apple comparison, we remain to train the same model of StyleGAN2 without decreasing its generative capacity by using fewer channels or shallower mapping networks since such designs require less data. Meanwhile, the generated resolution remains 256×256 and the datasets are amplified via the horizontal flip operation as well.

The quantitative and qualitative results are shown in Fig. 3b and Fig. 5 respectively. Obviously, FID significantly increases with the decreasing number of training images from 70K to 100. Nevertheless, our InsGen trained with only 100 unique images remains to outperform many approaches like PA-GAN in Fig. 3b with 2K images. Besides, with 500 training samples, our method is able to obtain the competitive performance to those using 10k images. Namely, our InsGen could improve the data-efficiency by more than 20×. Qualitative results suggest that our approach still produces meaningful images without incurring the model collapse no matter how many training images exist in the data collection.

## 5    Conclusion and Discussion

In this work, we develop a novel data-efficient Instance Generation (*InsGen*) method for training GANs with limited data. With the instance discrimination as an auxiliary task, our method makes the best use of both real and fake images to train the discriminator. In turn the discriminator is exploited to train the generator to synthesize as many diverse images as possible. Experiments under different data regimes show that InsGen brings a substantial improvement over the baseline in terms of both image quality and image diversity, and outperforms previous data augmentation algorithms by a large margin.

Although InsGen significantly improves the data efficiency in training generative models, it leaves some future work to do. One limitation of InsGen is that the performance gain becomes marginal when the training dataset is sufficiently large. This suggests that the discriminator can not benefit from the newly introduced instance discrimination any more. It may require a more challenging task to further improve the performance. Another limitation is that the FID score remains unsatisfying when the training data is extremely limited, say several hundred. It is worth exploring how to fully utilize the fake samples for discriminator training.

**Acknowledgments.** The project was supported through the Research Grants Council (RGC) of Hong Kong under ECS Grant No.24206219, GRF Grant No.14204521, CUHK FoE RSFS Grant.

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
