# More qualitative results of Data-Efficient Instance Generation from Instance Discrimination

We present more synthesized images of FFHQ with less training images in Fig. A3 and AFHQ in Fig. A1. Moreover, we qualitatively compare against ADA [1] in Fig. A2. Even if the number of training image becomes 2000, our approach remains to produce the photo-realistic images while the artifacts appear on ADA [1].

Cat (5153), FID 2.60

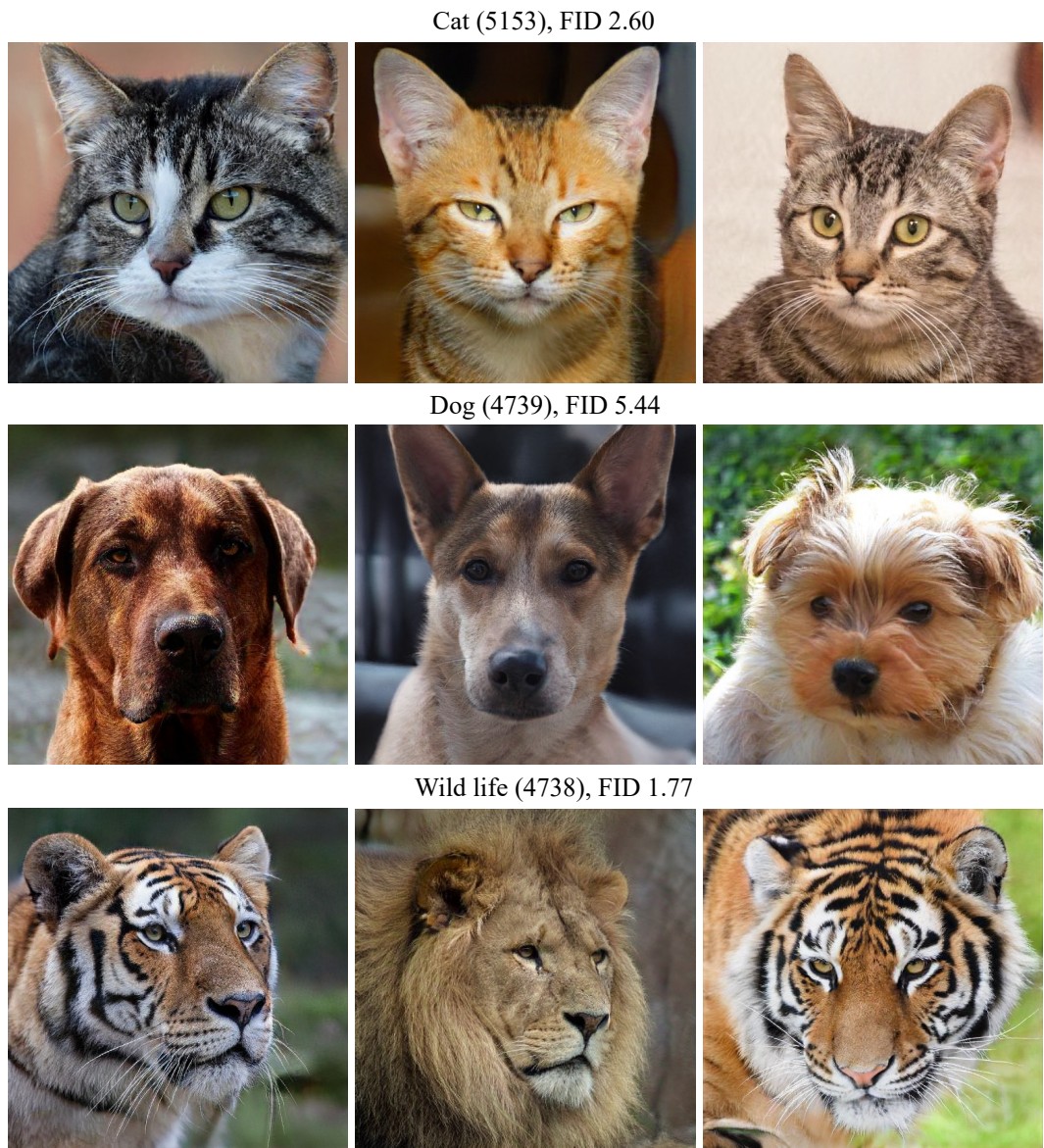

Dog (4739), FID 5.44

Wild life (4738), FID 1.77

Figure A1: Synthesized samples on AFHQ. All images are generated without truncation.

## References

[1] T. Karras, M. Aittala, J. Hellsten, S. Laine, J. Lehtinen, and T. Aila. Training generative adversarial networks with limited data. In *Adv. Neural Inform. Process. Syst.*, 2020. 1, 2

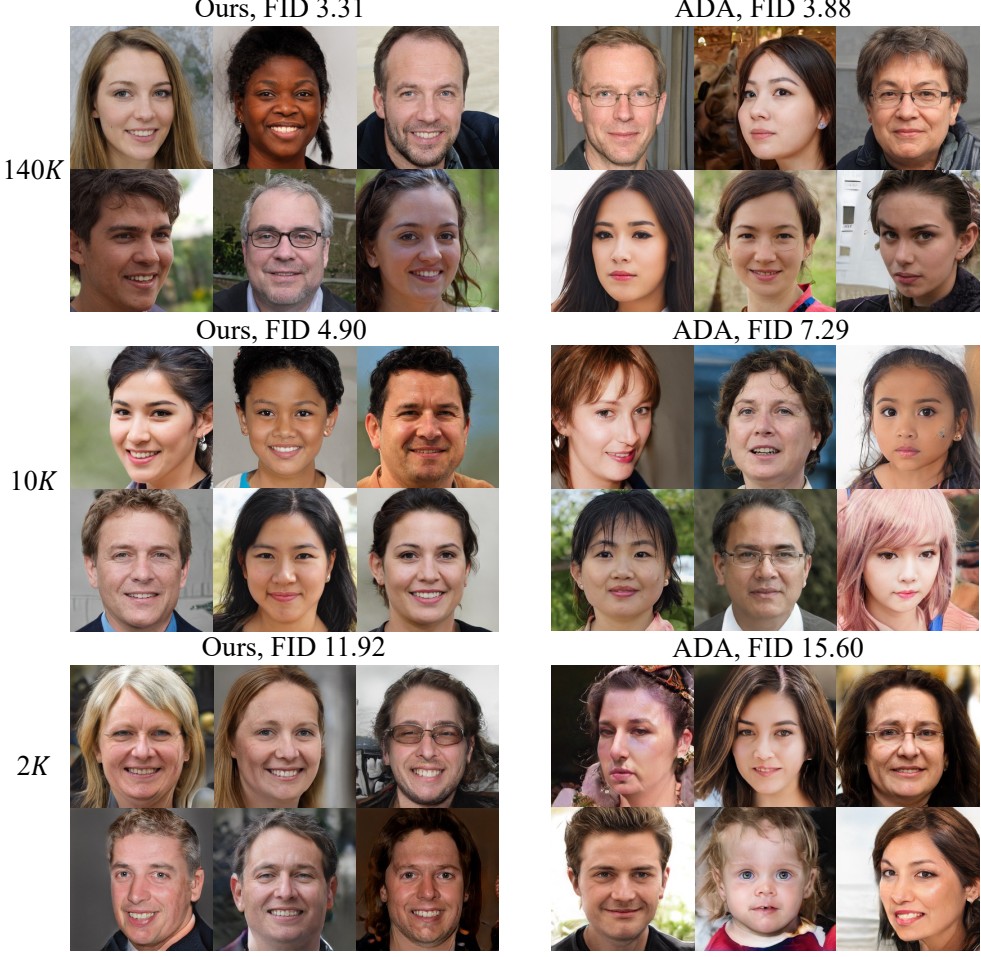

Figure A2: Qualitative comparison with ADA [1]. All images are generated with truncation.

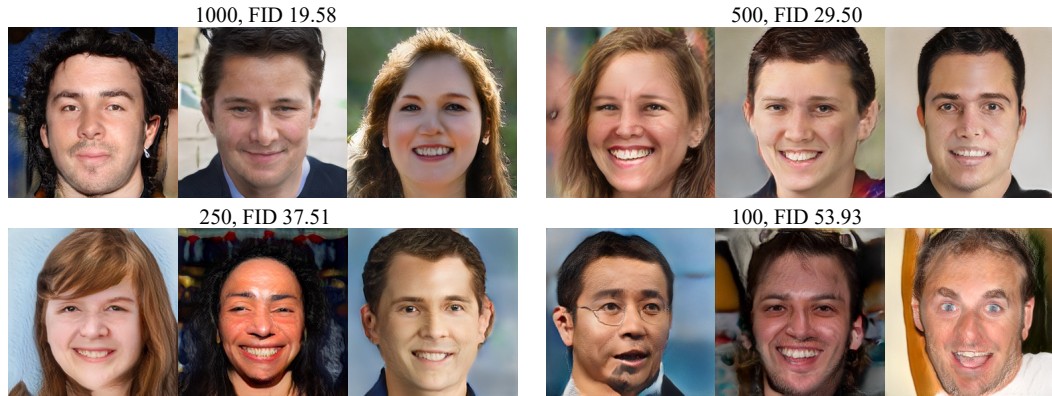

Figure A3: Synthesized samples on FFHQ with limited training data. All images are generated with truncation.