# OpenReview forum: "Data-Efficient Instance Generation from Instance Discrimination"
_NeurIPS.cc/2021/Conference — NeurIPS 2021 Poster_

### Official Review · Reviewer_eSj9 · 2021-06-29

**Rating:** 6
**Confidence:** 4

**Summary:**

This work adds an auxiliary contrastive learning task to train GANs in the regime of limited data.
In particular, they add an instance discrimination task to the discriminator and update both, generator and discriminator weights based on the modified objective.

**Ethical Concerns:**

Ethical concerns are not discussed in this work. However, there are important issues that should be discussed at least shortly. In particular because this work improves image synthesis *in the low-data regime*.

**Limitations And Societal Impact:**

As mentioned above, please discuss the limitations of your work, in particular, wrt training time and memory consumption. Also please discuss the societal impact.

**Main Review:**

Originality:
The paper integrates well-known ideas from contrastive learning into GAN training to improve performance, particularly in the low-data regime. While the idea is simple, combining both fields is a very interesting approach. The related work is well-written and gives the reviewer a good overview of the current state of the field and this work's contribution.

Quality:
Overall, the claims made in this paper are well supported, e.g. by the ablation in Section 4.2. However, the experimental evaluation could be more thorough:
   - To support the generalizability of this approach it would be good to extend the evaluation to a second architecture, e.g. https://arxiv.org/abs/2101.04775.
   - Please discuss the following limitations:
     i ) how does the contrastive objective impact the training time? Since the loss is computed over various samples does this slow down training or increase memory significantly?
     ii) could this additional objective prevent the GAN from converging to the optimal solution? How important is the weighting of the losses and how are the hyperparameters from the appendix selected?
    - Which choices of augmentations are most effective? Are the same augmentations needed as for ADA or are some less effective for the task of instance discrimination?
   - Additional ablations that should be discussed:
      i) If more synthetic samples keep improving FID (Figure 3), why stop at 1k? If feasible please add more results or discuss why this is not feasible in practice.
     ii) What is the value of epsilon (missing in hyper-parameters in the appendix) and how sensitive is the approach to this choice? Can you include some samples within the range of epsilon to give the reader an intuition on how similar these samples are?
      iii) Did you try using the same head for real/fake images (phi^r, phi^f)? While the given intuition (L.163) is comprehensible, an ablation study in the appendix would be good.
      iv) Why is the noise perturbation not applied to the generator? You could either give an intuition or add results of this ablation to the Table 3.
     v) Why do you use a momentum encoder? How much does this alone affect the results? Is this the standard (e.g. does ADA also use it?) or sth that you propose to do in this work?

Clarity:
The paper is well-written and well-structured. In particular, I like that it is self-contained and that the related work gives a good overview of the field.
Minor: L.207, L_D^r and L_D^f are not defined previously

Significance:
The results from this work might be useful to the community because it proposes a relatively simple method to improve performance in the low-data regime. To judge the practical value of the proposed method, it would be good to include an analysis on the overhead wrt training time / memory consumption regarding the added auxiliary task.

**Time Spent Reviewing:**

3

---

> ### Author Response · Authors · 2021-08-10
> **Response to Reviewer eSj9**
>
> Thanks for the valuable comments. Individual concerns are addressed as follows.
>
> **Q1. Extend the evaluation to a second architecture.**
>
> Please see **Results on SNDCGAN** in the Common Concerns.
>
> **Q2. How does the contrastive objective impact the training time? Since the loss is computed over various samples does this slow down training or increase memory significantly.**
>
> We only involve two heads on top of the original discriminator, which barely affect the computational cost. In fact, for the models training with $256\times256$ resolution, the training speed slows down 11%, and the memory of CPU and GPU increase 13% and 2.5% respectively. We believe that such a source requirement is acceptable for the exchange of significantly improved data efficiency. We will report the training time and the memory cost.
>
> **Q3. Could this additional objective prevent the GAN from converging to the optimal solution? How important is the weighting of the losses and how are the hyperparameters from the appendix selected.**
>
> Empirically, our method converges well in all experiments and stabilizes the training to some extent by observing the training curves (see supplementary material). Also, we do not pour much attention on the loss weight search and all experiments are conducted with the same loss weights (as introduced in the supplementary material). It suggests that our approach can be easily transferred to various settings with minor effort.
>
> **Q4. Which choices of augmentations are most effective? Are the same augmentations needed as for ADA or are some less effective for the task of instance discrimination.**
>
> Our main scope focuses on improving the data efficiency for image generation. For a fair comparison, as we mentioned in L198, we directly *reuse* the augmentation pipeline of StyleGAN2-ADA [24]. Which types of augmentations are effective is worth exploring. We leave it to future work.
>
> **Q5. If more synthetic samples keep improving FID (Figure 3), why stop at 1k? If feasible please add more results or discuss why this is not feasible in practice.**
>
> When we keep increasing the number of negative samples in Fig. 3, the FID would not decrease anymore (or even increase). For example, when we use $2K$ samples, FID becomes 12.14 while $1K$ leads to 11.92. One possible reason is that along with the size of the queue increasing, there may be some images that are very close to the anchor. Under such a case, treating all the images in the queue as negative samples (as proposed in MoCo) does not stand any longer. Also, it will increase the computational cost. Hence, we stop at using a queue with size $1K$.
>
> **Q6. What is the value of epsilon (missing in hyper-parameters in the appendix) and how sensitive is the approach to this choice. Including some samples within the range of epsilon to give the reader an intuition on how similar these samples are.**
>
> Intuitively, a too small epsilon will have no effect on the output image. On the other hand, the image synthesized with a too large epsilon cannot be viewed as a positive sample anymore. The key of contrastive learning is to find reasonable but challenging positive pairs and negative pairs. In our experiments, we use 0.15 for the noisy perturbation. We will discuss this and include the ablation study.
>
> **Q7. Did you try using the same head for real/fake images? An ablation study would be better.**
>
> Yes. If we use the same head for real/fake images, we would obtain 4.21 FID with $140K$ training images, while two heads lead to 3.31. Intuitively, the generator could not produce photo-realistic outputs at the early stage. If real and fake images share the same head, it may be hard for the discriminator to produce reliable features for contrastive learning. We will discuss this in revision.
>
> **Q8. Why is the noise perturbation not applied to the generator? You could either give an intuition or add results of this ablation to Table 3.**
>
> The motivation of noisy perturbation is to produce the challenging positive pair for contrastive learning. Due to the continuity of the GAN latent space, latent codes located within a small area are commonly assumed to be similar to each other, and hence should be projected to similar features under the view of the discriminator. However, the generator should be encouraged for diverse generation, which means that all samples should be different from each other, regardless of the distance between latent codes. Thus, the noise perturbation is not applied to the generator training.
>
> **Q9. Why do you use a momentum encoder? How much does this alone affect the results? Is this the standard (e.g. does ADA also use it?) or sth that you propose to do in this work?**
>
> Momentum encoder is proposed in MoCo-v2 [8], which is a necessary component for contrastive learning. It basically averages the parameters of the network to make sure that the features in the queue do not change too fast.
>
> **Q10. L_D^r and L_D^f are not defined previously.**
>
> Thanks. $L_D^r$ and $L_D^f$ should be $C_D^r$ and $C_D^f$. We will correct them in the next version.

---

> > ### Comment · Reviewer_eSj9 · 2021-08-30
> > **Response to Rebuttal**
> >
> > Dear authors,
> >
> > thank you for responding to my concerns. I generally find your approach interesting and appreciate the additional experiments and explanations you provide. However, I agree with reviewer 9ouC that your work could greatly benefit from an ablation on different objectives to study if the choice of objective indeed impacts the performance. In particular, the difference between using two different heads vs a shared head could suggest that the performance also varies between MoCO-v2 and SimCLR. Therefore, I decided to stick to my rating for now.

---

> > > ### Author Response · Authors · 2021-08-31
> > > **Response to Reviewer eSj9**
> > >
> > > Dear reviewer,
> > >
> > > Thanks for your feedback. The main difference between SimCLR and MoCo is the source of negative samples. Specifically, SimCLR takes an online strategy by regarding the other samples in the current data batch as the negative while MoCo uses a memory queue to store negative features. In our case, MoCo with a memory queue is much more flexible since we would like to leverage the infinite synthesized samples to alleviate the overfitting problem caused by insufficient training data. Our experimental results (L305) also suggest that involving more negative samples is of great benefit to the synthesis, especially with the limited training data. However, for SimCLR, we have to enlarge the total batch size to introduce more negative samples, which increases the computational complexity significantly.
> > >
> > > Due to the limited time, we will introduce SimCLR to InsGen and make a detailed comparison in the next version. Thanks for pointing this out.

---

> > > ### Author Response · Authors · 2021-09-02
> > > **Response to Reviewer eSj9**
> > >
> > > In **Common Concerns on Different Contrastive Methods**, we present the comparison between different contrastive objectives. The ablation on using SimCLR *v.s.* MoCo is also present.

---

### Official Review · Reviewer_JJnj · 2021-07-12

**Rating:** 6
**Confidence:** 4

**Summary:**

The paper proposes a kind of contrastive loss that aims to discriminate both real and generated instances for GAN training. Experiments on FFHQ and AFHQ demonstrate that the proposed method implemented upon StyleGAN2-ADA can improve FID especially with limited data.

**Limitations And Societal Impact:**

Good

**Main Review:**

Strengths:
- The proposed method sounds novel and well motivated.
- The proposed method can improve StyleGAN2-ADA's FID especially with limited data. It is good to see that the proposed method can also improve its FID on the full FFHQ training set.

Weaknesses:
- My main concern is that the authors did not provide any empirical evidence on the generator diversity as the authors claimed. With limited data, the generator can memorize the full training set very easily, i.e., the generator may simply produce identical training images and thus achieve a low FID. Thus it is necessary to conduct nearest neighbor tests, style space interpolation, or measure the diversity quantitatively using e.g. precision/recall.
- I think the proposed loss combination is a bit complex but not fully justified. For example, why can't we simply apply data augmentation to fake images as done in real images instead of noise perturbation?
- The experiments are limited to the StyleGAN2-ADA backbone, so it is not clear whether the proposed method can generalize over other GAN architectures/algorithms. The datasets are also a bit limited.
- In the supplementary material, only the fact that the discriminator achieves better accuracy on the training set is not meaningful.

====== Post Rebuttal ======

Changed to 6 provided that the authors have promised to include diversity experiements in the revision and addressed my remaining concerns.

**Time Spent Reviewing:**

3

---

> ### Author Response · Authors · 2021-08-10
> **Response to Reviewer JJnj**
>
> Thanks for the valuable comments. Individual concerns are addressed as follows.
>
> **Q1. No empirical evidence on the generator diversity as the authors claimed. it is necessary to conduct nearest neighbor tests, style space interpolation, or measure the diversity quantitatively using e.g. precision/recall.**
>
> Thanks for the suggestion on the diversity experiment. We follow ADA [24] to evaluate the diversity of different models trained on FFHQ-256 with the precision and recall metrics, beyond FID. Intuitively, a higher precision means more realistic images while higher recall indicates larger amount of variation. Results are shown as follows:
>
> | Models | #Samples | FID↓ | Precision↑ | Recall↑ |
> | :-- | :-: | :-: | :-: | :-: |
> | ADA [24]      | $140K$ | 3.88 | 0.680 | 0.433 |
> | InsGen (ours) | $140K$ | **3.31** | **0.688** | **0.472** |
> | ADA [24]      |  $10K$ | 7.29 | 0.694 | 0.269 |
> | InsGen (ours) |  $10K$ | **4.90** | **0.699** | **0.339** |
> | ADA [24]      |   $2K$ | 15.6 | 0.670 | 0.104 |
> | InsGen (ours) |   $2K$ | **11.9** | **0.674** | **0.201** |
>
> Obviously, our InsGen improves the synthesis from the perspective of both the realistic quality and the image diversity. We will include the above analysis and also involve the nearest neighbor tests and style space interpolation in the next version.
>
> **Q2. I think the proposed loss combination is a bit complex but not fully justified. For example, why can't we simply apply data augmentation to fake images as done in real images instead of noise perturbation.**
>
> As mentioned in Eq. (6), we do apply data augmentations and noisy perturbation to fake images at the same time.
>
> Recall that the only type of loss (beyond the adversarial loss) used in InsGen is the contrastive loss as in Eq. (4). This loss is used on real images, fake images (with noise perturbation), and the training of the generator. All these terms are studied in the ablation study (see Tab. 3). The loss weights to balance these terms are also introduced in the supplementary material, which we will move to the main paper in the next version.
>
> **Q3. Generalize over other GAN architectures/algorithms/dataset.**
>
> Please see **Results on CIFAR-10** and **Results on SNDCGAN** in the Common Concerns.
>
> **Q4. In the supplementary material, only the fact that the discriminator achieves better accuracy on the training set is not meaningful.**
>
> The experiment is conducted with $2K$ training images from FFHQ. This gives us a chance to evaluate the discriminator with the remaining real samples. Specifically, we re-evaluate models on the 50000 real images from the remaining set and 50000 generated images, and then calculate the predicted score (*i.e.*, the output of the discriminator) from those images. Note that a larger score means the image is more realistic under the view of the discriminator. It turns out that InsGen has 2.02 and -1.93 for real and fake data, while ADA has 1.13 and -1.06. It suggests that the discriminator has a more powerful discriminative ability after being equipped with the instance discrimination task.

---

> > ### Comment · Reviewer_JJnj · 2021-08-29
> > **Reply**
> >
> > Thanks for the response. Nice to hear that the proposed method can improve diversity. This is important especially in few-shot tasks.

---

### Official Review · Reviewer_HMNP · 2021-07-13

**Rating:** 7
**Confidence:** 5

**Summary:**

This paper introduces a novel GAN training method, named InsGen. This work effectively overcomes the performance degradation problem of GAN training with a small amount of the train dataset. The main idea is to train a discriminator by assigning an instance discrimination task as well as a bi-classification task (e.g. real or fake). The experimental results show that, with the smaller amount of data used, the synthesized image quality of InsGen outperforms the images from SOTA methods.


**Limitations And Societal Impact:**

Yes

**Main Review:**

Strength
* This paper is well written and easy to follow up.
* The idea of noise perturbation is novel (and clever!).
* The idea of enhancing discriminator by the instance discrimination does help enhance overall generation quality of GANs.
* Experimental results show a significant improvement when training GANs using a small amount of data.

Weakness
* Adding the experiments using datasets with various classes would be more convincing. At least, the experiment on cifar-10 should be appended.
* This method does not seem to require a specific model structure, isn't it? It seems that the training method of InsGen can be applied not only to StyleGAN v2 but also to other structures such as BigGAN. However, in the current information, it is not clear. (If the method does not require specific model architectures, the authors should advertise themselves.)
* According to Tab.3, the 70K FID of the 3rd-row increases (which means quality degradation) compared to the 2nd-row. Any explanation?

**Time Spent Reviewing:**

5

---

> ### Author Response · Authors · 2021-08-10
> **Response to Reviewer HMNP**
>
> Thanks for the valuable comments. Individual concerns are addressed as follows.
>
> **Q1. Adding the experiments using datasets with various classes would be more convincing. At least, the experiment on cifar-10 should be appended.**
>
> Thanks. Please see the conditional setting in **Results on CIFAR-10** in the Common Concerns.
>
> **Q2. Whether the specific model structure is required.**
>
> InsGen does not rely on the model structure. Please see **Results on SNDCGAN** in the Common Concerns for performance comparison on a second network architecture, SNDCGAN, beyond StyleGAN2.
>
> **Q3. According to Table 3, the 70K FID of the 3rd-row increases (which means quality degradation) compared to the 2nd-row.**
>
> When the training set is sufficiently large (*e.g.*, having 70K samples), the task of real image discrimination is already challenging enough. Under such a case, the discriminator can barely benefit from distinguishing fake images, given the vanilla augmentation approaches. Or, it may need careful design to balance different loss terms, which we do not put much effort into. Instead, we propose a noise perturbation strategy to produce positive pairs for fake image discrimination. After introducing such a strategy, the FID score decreases again. We will discuss this.

---

### Official Review · Reviewer_9ouC · 2021-07-17

**Rating:** 6
**Confidence:** 5

**Summary:**

This paper propose to add instance discrimination (aka contrastive learning) objective for training generative adversarial networks. Specifically, (a) instance discrimination among real samples & (b) instance discrimination among fake samples are used as two axillary objectives. In addition, noise perturbation whereby the method uses z & z+epsilon to create two views needed for contrastive learning. (conventional contrastive learning uses 2 different augmentations of an image to create 2 views). Experimental results show that the proposed method InsGen outperforms prior methods that use data augmentations for training image GANs.

**Limitations And Societal Impact:**

Limitations or potential negative societal impact are not included in this paper. Including limitations as well as the position of this paper against prior literature will greatly enhance the quality of this paper.

**Main Review:**

[Strengths]
- Overall, the paper is well-written and easy to follow.
- Using noise perturbation to create two views is pretty novel.
- The method can have high impact to the field, as it is simple but highly effective. I think it is much easier to use compared to ContraD --- which is another method that used contrastive learning for GAN training. Personally, I have tried using ContraD for my project, but it was not easy to implement. This method can be a good alternative to ContraD.

[Weaknesses]
- Novelty and Comparison against prior art: There is prior works in GAN literature that used instance discrimination objectives: ContraD [20] and DC-VAE [a]. Conceptually, InsGen is pretty similar to these works.
I believe comparison against these works should be written explicit in the paper.
This submission (InsGen) has subtle differences against ContraD:
(1) ContraD does not use instance discrimination for training the generator. InsGen does.
(2) A novel way of creating two views (noise perturbation).
(3) ContraD uses SimCLR (for real samples) & SupCon (for fake samples), whereas InsGen uses MoCo.

I think (1) & (2) are addressed in ablation, but (3) is not well-studied. The effect of different contrastive learning methods should be investigated. It is possible that the performance difference between ContraD and InsGen is due to the difference between SimCLR and MoCo.

Also, Table 1 (FFHQ) does not include ContraD. ContraD should be compared.

- Missing details about augmentation. What kinds and strengths of data augmentation operations are used for MoCo? Does it follow the original setting of MoCoV2?

- Missing results on CIFAR10. Since CIFAR10 is the most widely studied benchmark for Image GANs, it is better to include it.

[Missing References]
[a] Gaurav Parmar, Dacheng Li, Kwonjoon Lee, and Zhuowen Tu. Dual contradistinctive generative autoencoder. In CVPR, 2021.

======= Post-Rebuttal ======
After reading the latest results, I increased my rating from 5 to 6 for following reasons:
(a) the improvement is due to the difference in formulation (not the underlying contrastive learning method);
(b) this method is cleaner and easier to use than ContraD.
The formulation of ContraD is not natural and kind of hard to implement (from my experience). Although ContraD is a similar work on introducing contrastive learning to the GAN training, I believe the community will benefit from easier-to-use and better performing variant. Hence, I vote for accepting this paper.

**Time Spent Reviewing:**

1.5

---

> ### Author Response · Authors · 2021-08-10
> **Response to Reviewer 9ouC**
>
> Thanks for the valuable comments. Individual concerns are addressed as follows.
>
> **Q1. The main differences from ContraD.**
>
> InsGen differs from ContraD in the following aspects:
>
> 1. We have a different usage of fake images. Given a synthesized sample, ContraD treats other synthesized samples as positive, still leaving the discriminator with a bi-classification task. By contrast, InsGen asks the discriminator to also distinguish every individual fake image, leading to a more powerful discriminator especially when the real data is limited. This contribution is crucial to the low-data setting because we can utilize an unlimited number of fake images to improve the generator.
> 2. We train the generator in different manners. In ContraD, the generator is trained to fool the discriminator, which is the same as most existing GAN variants. Beyond the task to fool the discriminator, InsGen trains the generator explicitly with the *newly introduced* instance discrimination task. This greatly improves the diverse generation of the generator under the low-data setting.
> 3. We use different ways to produce positive pairs for contrastive learning. ContraD uses the conventional data augmentation pipeline, while InsGen proposes noise perturbation strategy which leverages the continuity of latent space to produce hard samples for instance discrimination task.
> 4. InsGen shows better performance and stronger data efficiency than ContraD, which is suggested on AFHQ and CelebA-HQ-128 as shown in the common concerns.
>
> **Q2. Effect of different contrastive loss.**
>
> Indeed, the way to implement contrastive loss may affect the results. But, as illustrated in **Q1**, we differ from ContraD far beyond merely changing SimCLR to MoCo-v2 [8]. With different formulations between ContraD and InsGen, it becomes difficult to make an apple-to-apple comparison. On the other hand, MoCo-v2 and SimCLR achieve similar performance in representation learning, therefore, we believe that the implementation of contrastive loss is not the main reason that InsGen outperforms ContraD. But, as suggested, it will be of great help to also investigate the effect of different contrastive losses (*e.g.*, SimCLR, SwAV, BYOL, *etc.*) in our framework. We leave it to future work.
>
> **Q3. Compare with ContraD on FFHQ.**
>
> ContraD did not conduct experiments on FFHQ. But we compare with it on AFHQ for high-resolution image generation (see Table 2 of the submission), and on CelebA-HQ-128 for face generation (see **Results on SNDCGAN** in the Common Concerns). We surpass ContraD in all experimental settings.
>
> **Q4. Missing details about augmentations.**
>
> As mentioned in L198, we *reuse* the augmentation pipeline (including augmentation types and strengths) of StyleGAN2-ADA [24] for a fair comparison. We will highlight it in the next version.
>
> **Q5. Missing references.**
>
> Thanks. We will add the missing references.
>
> **Q6. limitations or potential negative societal impact are not included in this paper.**
>
> One limitation of InsGen is that the performance gain becomes marginal when the training dataset is sufficiently large. This suggests that the discriminator can not benefit from the newly introduced instance discrimination any more. It may require a more challenging task to further improve the performance. Another limitation is that the FID score remains unsatisfying when the training data is extremely limited, say several hundred. It is worth exploring how to fully utilize the fake samples for discriminator training.
>
> As for the social impact, our approach makes it possible to train a high-fidelity generative model with limited data, significantly lowering the bar for producing fake data. We will discuss this in the next version.

---

> > ### Comment · Reviewer_9ouC · 2021-08-31
> > **Response**
> >
> > Thank you for the rebuttal.
> >
> > I am well aware of the difference of this formulation against ContraD. I have implemented ContraD and some variants to train my own GANs, so I am familiar with the setting. I still think the difference between this submission and ContraD is really small and subtle. Based on my experience, type of contrastive loss used (SimCLR, MoCo, etc) has a huge impact on the final performance. This submission presents an ablation on exact formulation of contrastive loss, but similar kind of ablation is also presented in ContraD.
> >
> > Before seeing the effect of contrastive loss (SimCLR vs MoCo), I cannot feel confident about voting for acceptance. Hence, I maintain my original rating.

---

> > > ### Author Response · Authors · 2021-09-02
> > > **Response**
> > >
> > > The formulation of our InsGen is fundamentally different from ContraD, not only by using a second contrastive method. Following your suggestion, we replace MoCo with SimCLR in InsGen and also surpass ContraD by a large margin. Please refer to the **Common Concerns on Different Contrastive Methods** for detailed comparison.

---

> > > > ### Comment · Reviewer_9ouC · 2021-09-03
> > > > **Updated Score**
> > > >
> > > > Thanks for the effort on extra experiments. This resolves my major concern. I changed my rating from 5 to 6.

---

### Author Response · Authors · 2021-08-10
**Common Concerns (Rebuttal)**

This work focuses on data-efficient image generation and is built on the state-of-the-art framework, *i.e.*, StyleGAN2-ADA [24]. Experimental results suggest that our InsGen surpasses such a **strong baseline** by a large margin, especially when the data is limited. Indeed, as pointed out by the reviewers, this work can *be beneficial from demonstrating its generalization to more datasets and network structures*. Hence, we would like to first show some results by applying InsGen to the CIFAR-10 dataset and the SNDCGAN framework. All the results will be added to the revised version.

### Results on CIFAR-10

We conduct experiments on CIFAR-10 with **both the unconditional setting and the conditional setting**. Concretely, the unconditional setting ignores the label and treats all samples equally, while the conditional setting aims at learning a category-aware image synthesis model. We still choose StyleGAN2-ADA [24] as our **strong baseline** and strictly follow the official implementation and training details. Comparison results are shown below:

| Unconditional Setting | | | |
| :-- | :-: | :-: | :-: |
| Method                 | Seen Images | FID↓ |  IS↑ |
| ProGAN in [24]         |        100M | 15.52| 8.56 |
| AutoGAN in [24]        |        100M | 12.42| 8.55 |
| ADA in [24]            |        100M | 2.92 | 9.83 |
| ADA (our reproduction) |         50M | 3.01 | 9.78 |
| InsGen (ours)          |         50M | **2.70** | **9.92** |

| Conditional Setting | | | |
| :-- | :-: | :-: | :-: |
| Method                 | Seen Images | FID↓ |   IS↑ |
| BigGAN in [24]         |        100M | 8.47 | 9.07  |
| MultiHinge in [24]     |        100M | 6.40 | 9.58  |
| ADA in [24]            |        100M | 2.42 | 10.14 |
| ADA (our reproduction) |         50M | 2.62 |  9.91 |
| InsGen (ours)          |         50M | **2.24** | **10.28** |

Notably, the original paper [24] reports the results after the discriminator has seen 100M images. Due to the limited time, however, we report the results after the discriminator has seen 50M images. We can tell that InsGen performs better than the baseline model given the same seen images, and *even* beats the 100M model. We also borrow the results of other alternatives from ADA [24], *e.g.*, ProGAN and AutoGAN for the unconditional generation, BigGAN and MultiHinge for the conditional generation. We can see that their performances are much worse than StyleGAN2-ADA [24], **which is why we choose ADA as our strong baseline**. We would like to show that **InsGen can help improve the data efficiency in GAN training even when the base model is already powerful enough**.

### Results on SNDCGAN

To further verify the generalizability of InsGen, we choose another framework, SNDCGAN (which introduces spectral normalization into DCGAN), as the baseline like ContraD [20] has done. We use CelebA-HQ-128 for the fair comparison with ContraD and other alternatives. Here, we strictly follow the training hyper-parameters (*e.g.*, number of iterations, batch size, learning rate, training and validation set separation, *etc.*) reported in ContraD [20]. The results are shown below, where all numbers except InsGen are borrowed from [20].

| Method | FID↓ |
| :-- | :-: |
| SNDCGAN [20]  | 24.8 |
| CR [20]       | 20.9 |
| bCR [20]      | 19.5 |
| ContraD [20]  | 17.1 |
| InsGen (ours) | **15.6** |

The results suggest that our InsGen achieves the best performance compared to existing approaches. It demonstrates that **InsGen does not necessarily rely on the network architecture and can be easily transferred to other models with minor effort**.

Till now, we have demonstrated the generalization of InsGen to more datasets (unconditional CIFAR-10, conditional CIFAR-10, CelebA-HQ-128) and the SNDCGAN architecture. We will respond to the concerns from each reviewer separately.

---

### Author Response · Authors · 2021-09-02
**Common Concerns on Different Contrastive Methods**

During the discussion, two reviewers raised the concern on how different contrastive methods (*e.g.*, MoCo, SimCLR, *etc.*) may affect the performance. We would like to address this concern with some experimental proof.

First of all, we would like to reaffirm that **besides using different contrastive methods, our InsGen differs from ContraD [20] fundamentally, as we have a different formulation**. Indeed, we both introduce contrastive learning into the training of GANs. However, there are still many differences between these two approaches. We list some major ones as follows.

- ContraD introduces a separate head for the conventional adversarial loss (*i.e.*, real *v.s.* fake), such that the adversarial loss is only back-propagated to the head but not the backbone. In other words, the backbone is only trained with the contrastive loss. We argue that such a formulation may lead the discriminator to ignore some realistic-related information and hence fail to provide enough guidance to the generator. Differently, we keep the original bi-classification task of the discriminator and **introduce contrastive learning as a new one**. In this way, we expect the discriminator backbone to produce more discriminative features regarding both tasks. From the comparisons on CIFAR-10 and SNDCGAN, we can tell that our formulation indeed gives better performances.

- When applying contrastive learning to fake samples (synthesized by the generator), ContraD treats all fake samples as the same class, and all real samples as the other one. In other words, this part is very similar to conventional adversarial loss. From this perspective, the introduced contrastive learning on fake samples may have marginal effect compared to the original GAN. Differently, we **treat each synthesized sample as an independent instance**. In this way, we are use "infinite" fake data for the training of the discriminator, enhancing its discriminative power **especially when the real data is limited**.

- In ContraD, each fake sample only has one "view" (following the description in contrastive learning). We introduce "noise perturbation" to create different views of a fake sample. Such a perturbation is different from the image-based augmentations (like crop, resize, color jittering, *etc.*), which are applied onto real samples, and hence increases the difficulty of the instance discrimination task. In this way, the discriminator can benefit from the more challenging task.

- In ContraD, contrastive loss is only used for discriminator training. But in our InsGen, the generator is also asked to make every synthesized instance to be different from each other. This improves the generation diversity (see **Q1 to Reviewer JJnj**).

We will add the above discussion into Relate Work to make the comparison clear in the revised version. Thanks for pointing this out.

Now, we show some experimental results by incorporating InsGen with SimCLR (instead of MoCo) as requested. Due to the limited resources, we only conduct experiments on AFHQ-CAT. Here, we exactly follow the implementation of SimCLR.

| AFHQ-CAT | | |
| :-- | :-: | :-: |
| Method              | Seen Images | FID↓ |
| ContraD [20]        |         25M | 3.82 |
| InsGen-MoCo         |         10M | 2.60 |
| InsGen-SimCLR       |         10M | 2.84 |

We can tell that, after changing MoCo to SimCLR, our InsGen also surpasses ContraD on AFHQ-CAT dataset by a large margin, demonstrating that the improvement of InsGen over ContraD is not because of using MoCo instead of SimCLR. We will add the results.

We also notice that InsGen-MoCo is better than InsGen-SimCLR. That is because MoCo introduces a queue for computing contrastive loss, which means that MoCo uses more instances for instance discrimination at one time. This well meets our motivation, which is to use as many negative samples as possible to enhance the discriminative power of the discriminator. For the same purpose, SimCLR may require a much larger batch size to collect as many instances as possible. We believe that using SimCLR with a larger batch size may achieve the same performance as MoCo, but it may increase the computational complexity significantly. This is also the reason why we choose MoCo as our contrastive method. We will also explain this in the revised version.

---

### Decision · Program_Chairs · 2021-09-27

**Decision:**

Accept (Poster)

**Comment:**

All reviewers agree that the paper should be accepted, given it's effectiveness. I agree with the reviewers. The contents of the rebuttal are important to distinguish the paper from previous work (especially ContraD) and therefore should be incorporated into the final version.